# Length-dependent flagellar growth of *Vibrio alginolyticus* revealed by real time fluorescent imaging

**Meiting Chen[1†], Ziyi Zhao[2†], Jin Yang[2†], Kai Peng[2], Matthew AB Baker[3], Fan Bai[2*], Chien-Jung Lo[1*]**

[1]Department of Physics and Graduate Institute of Biophysics, National Central University, Jhongli, Taiwan; [2]Biodynamic Optical Imaging Center (BIOPIC), School of Life Sciences, Peking University, Beijing, China; [3]EMBL Australia Node for Single Molecule Science, University of New South Wales, Sydney, Australia

**Abstract** Bacterial flagella are extracellular filaments that drive swimming in bacteria. During motor assembly, flagellins are transported unfolded through the central channel in the flagellum to the growing tip. Here, we applied in vivo fluorescent imaging to monitor in real time the *Vibrio alginolyticus* polar flagella growth. The flagellar growth rate is found to be highly length-dependent. Initially, the flagellum grows at a constant rate (50 nm/min) when shorter than 1500 nm. The growth rate decays sharply when the flagellum grows longer, which decreases to ~9 nm/min at 7500 nm. We modeled flagellin transport inside the channel as a one-dimensional diffusive process with an injection force at its base. When the flagellum is short, its growth rate is determined by the loading speed at the base. Only when the flagellum grows longer does diffusion of flagellin become the rate-limiting step, dramatically reducing the growth rate. Our results shed new light on the dynamic building process of this complex extracellular structure.

**\*For correspondence:** fbai@pku.edu.cn (FB); cjlo@phy.ncu.edu.tw (C-JL)

[†]These authors contributed equally to this work

**Competing interests:** The authors declare that no competing interests exist.

## Introduction

Protein transport in biology can involve active transport by molecular machines and also passive transport through diffusion (*Rapoport, 2007*; *Yuan et al., 2010*; *Jacobson et al., 1987*). Although intracellular protein transportation has been intensively studied, extracellular protein transportation has yet to be thoroughly investigated (*Taschner and Lorentzen, 2016*; *Evans et al., 2013*). One particular puzzle in extracellular transport is how flagellin proteins are transported many microns and assembled to form the long extracellular organelle of the bacterial flagellum.

In aqueous solutions, most bacteria swim by rotating their flagella (*Berg, 2003*; *Sowa and Berry, 2008*). Using the transmembrane electrochemical proton motive force (PMF) to power the bacterial flagellar motor (BFM) (*Manson et al., 1977*; *Gabel and Berg, 2003*), fast rotating flagella can propel their cell body at a speed of 15–100 µm/s (*Xue et al., 2015*). Bacterial flagella, usually about 3–10 times the length of the cell body, are hollow protein cylinders of 20 nm outer diameter and 2 nm inner diameter (*Yonekura et al., 2003*). This long extracellular organelle is self-assembled from thousands of flagellin monomers (*Figure 1a*) (*Yonekura et al., 2003*; *Asakura, 1970*). Previous research has established that the flagellum assembles from the inside out, beginning with the basal body of the BFM, after which a type III flagellar secretion system (T3FSS) attaches to the base of this basal body (*Macnab, 2003*; *Minamino et al., 2008*; *Chevance and Hughes, 2008*). Utilizing energy derived from ATP hydrolysis and PMF (*Paul et al., 2008*; *Minamino and Namba, 2008*; *Lee and Rietsch, 2015*), the T3FSS continuously unfolds and delivers flagellin to the hollow interior of the motor, whereupon each unfolded flagellin protein is transported to the end of the filament and there

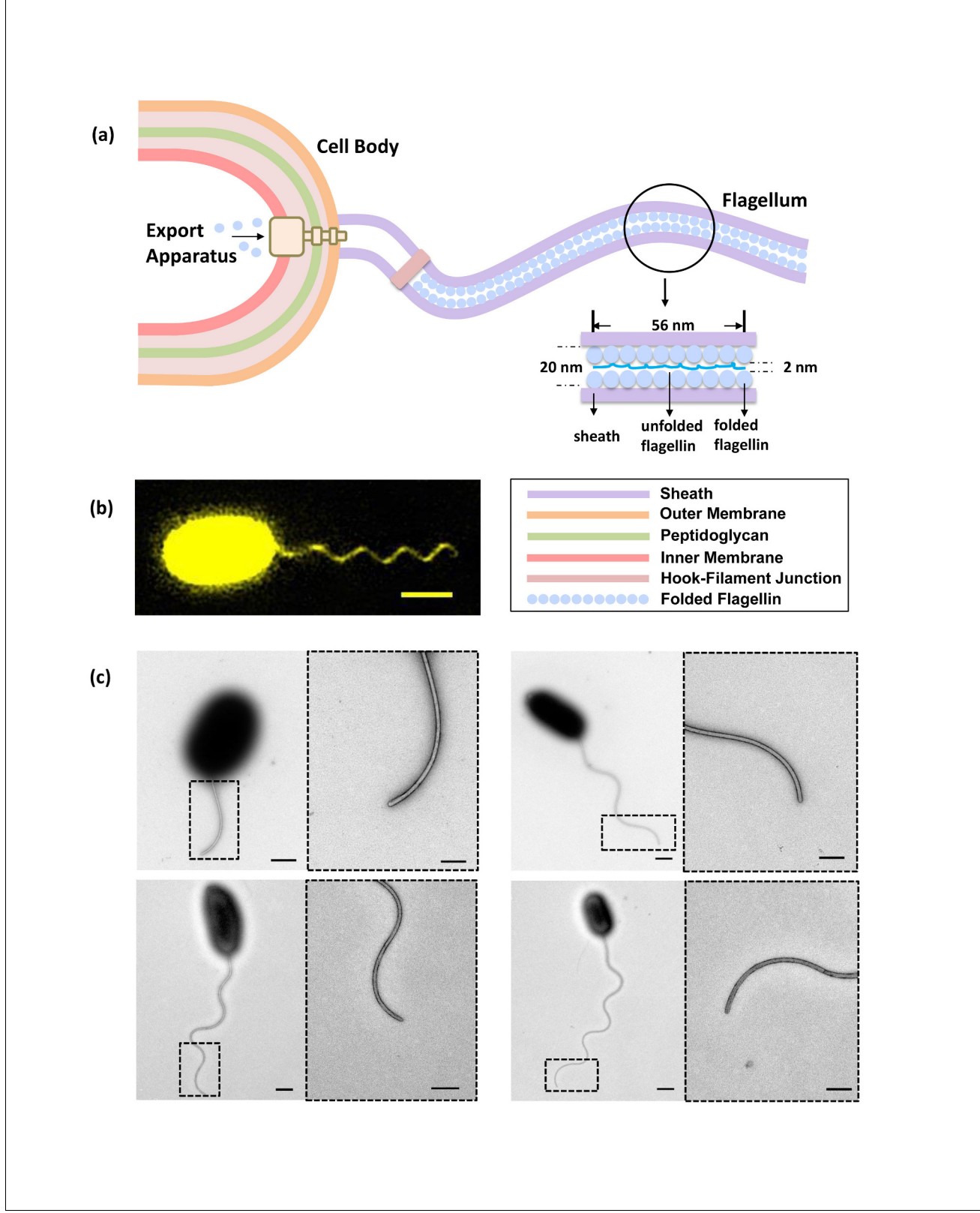

**Figure 1.** Bacterial flagellum of the *Vibrio alginolyticus*. (a) Schematic showing major components of the bacterial flagellar system. (b) Representative high-resolution fluorescence image of *Vibrio alginolyticus* polar flagellum taken by the Structured Illumination Microscopy (SIM, Materials and methods). Scale bar, 1.5 µm. (c) TEM images of cells with different lengths of flagella (1.8 µm, 4.5 µm, 5.3 µm and 7.8 µm, respectively), which are all tightly covered by sheath. Scale bars, 500 nm (200 nm in magnified area).

*Figure 1 continued on next page*

*Figure 1 continued*

The following figure supplements are available for figure 1:

**Figure supplement 1.** Negative-staining EM images of the sheathed polar flagella of *Vibrio alginolyticus*.

**Figure supplement 2.** *V.alginolyticus* flagellar length depends on culturing time.

**Figure supplement 3.** NanoOrange labeling does not affect cell viability or flagellar growth.

polymerizes to extend the structure (*Chevance and Hughes, 2008*; *Galán et al., 2014*; *Diepold and Armitage, 2015*).

Up till now, most studies have focused on the function of the flagellar motor (*Berg, 2003*; *Kojima and Blair, 2004*; *Lo et al., 2007*; *Stock et al., 2012*; *Hosu et al., 2016*; *Baker et al., 2016*) and molecular structure of the major components of the flagellar system (*Yonekura et al., 2003*; *Erhardt et al., 2010*; *Lee et al., 2010*; *Ibuki et al., 2011*), whilst there have been only limited efforts to characterize the dynamics of flagellar growth. In 1974, Iino (*Iino, 1974*) used electron microscopy to observe the lengths of *Salmonella typhimurium* flagella in different growth phases and reported that the flagellar growth rate decays exponentially with its length. Later, *Aizawa and Kubori (1998)* used dark-field microscopy to measure the bacterial flagellar length distribution at different time points and confirmed the same result. In contrast, more recently, (*Turner et al., 2000*, *2012*) labeled *Escherichia coli*'s pre-existing and newly grown flagellar segments with fluorescent dyes of two different colors and found that the average flagellar growth rate is a constant value independent of the flagellar length. There currently exist different theoretical predictions on the relationship between growth rate and length (*Keener, 2006*; *Schmitt and Stark, 2011*; *Tanner et al., 2011*; *Stern and Berg, 2013*). Despite this controversy, nearly all existing experimental investigations have been based on population measurements of flagellar length at two different time points at long intervals. For a better understanding of flagellar growth dynamics, a comprehensive, accurate and in vivo real-time measurement of single flagellar growth is required.

Here, we developed a fast and high-resolution fluorescence imaging method that enables in vivo real-time recording of *Vibrio alginolyticus* polar flagellar growth. We observed that the flagellar growth rate of *V. alginolyticus* is highly length-dependent, with distinct growth regimes. Initially, the flagellum grows at a constant rate when its length is below 1500 nm. While it continues to grow, the growth rate decreases sharply. To our knowledge, this is the first in vivo real-time recording of single filament growth, covering the full range of filament lengths. Our study reveals the details of how flagellar growth rate depends on its length and also provides a mathematical model that explains our major findings and reconciles previous observations.

## Results

### Fluorescent labeling of flagellar sheath for real-time monitoring of flagellar growth

Accurate real-time measurement of flagellar growth in live cells is challenging because (1) flagella are thin extracellular filaments and thus there is not the contrast to observe the flagellar growth under bright-field microscopy. Observation of flagella on a living cell can be achieved by dark-field microscopy, but its spatial resolution is very poor and only flagella longer than 4–5 μm can be resolved (*Aizawa and Kubori, 1998*) due to the strong light scattering from the cell body. Electron microscopy (EM) offers high enough resolution, but with that technology live cell imaging is not possible. Thus, we have to rely on fluorescent imaging to measure flagellar growth; (2) labeling must be quick to allow subsequent staining of newly grown flagellar filament. Also, to enable real-time measurement, the staining reagent has to be present in the bacterial growth medium. Therefore, the label must be non- or weakly fluorescent when not binding to its target, otherwise the background will be too high to resolve the growing filament; (3) most bacteria species possess multiple

extracellular flagella, all of which are swinging and can be entangled, making accurate measurement of single flagellar growth difficult.

To overcome these difficulties, we used *V. alginolyticus* as our experimental system, which has several advantages: (1) *V. alginolyticus* has only one polar flagellum (*Furuno et al., 1997*; *Zhu et al., 2013*), thus reducing flagella entangling and making it easier to track; (2) The polar flagellum is covered by sheath, which is believed to be a membrane-like structure that contains proteins, lipids and lipopolysaccharide and is contiguous with the outer membrane (*McCarter, 2001*). Previous research (*McCarter, 2001*; *Allen and Baumann, 1971*; *Glauert et al., 1963*) and our TEM studies revealed the synchronized growth of sheath and flagellum, with the sheath always tightly wrapping the inner flagellum (*Figure 1c*) and *Figure 1—figure supplements 1*; (3). The rich lipids constituent in the sheath can be easily labeled by a hydrophobic domain targeting fluorescent dye, NanoOrange (*Grossart et al., 2000*). NanoOrange is virtually non-fluorescent before binding and increases its fluorescence dramatically after binding to the sheath (*Grossart et al., 2000*). Therefore, high-resolution fluorescence images of flagella with good contrast can be obtained without washing away the labeling reagent (*Figure 1b*), which allows subsequent staining of newly grown segments; (4) The labeling speed of NanoOrange is very fast (<1 min) and does not influence flagellar growth (*Figure 1—figure supplement 3*), ensuring immediate visualization of newly grown flagellum, which is necessary to measure flagellar growth in real time; 5) Along with the cell body, the sheathed flagellum is prone to attach to the poly-L-lysine coated coverslip surface. This stabilizes the geometry of flagellar growth, making accurate measurement of flagellar length possible.

In our experiment, *V. alginolyticus* bacterial cells were grown following previously established protocols (Materials and methods). Before imaging, flagella labeling was performed by adding NanoOrange into the imaging medium. During imaging, NanoOrange was maintained in the medium to ensure timely labeling of newly grown flagellar segments. For the real-time observation of flagellar growth, cells attached to the coverslip were imaged under an automatic fluorescence microscope equipped with perfect focus system (Materials and methods). *Figure 2a* shows representative time-lapse images of the fluorescent flagellum of the same cell at time T and T+$\Delta$T, with the lengths L and L+$\Delta$L, respectively. Since the length we could measure is the 2D projection of the 3D flagellar length on the imaging plane (*Figure 2b*), the $L_{3D}$ can be calculated from $L_{2D}$ (*Figure 2—figure supplement 1*). Therefore, the instantaneous growth rate of the flagellum is defined as $\Delta L_{3D}/\Delta T$ and the value is assigned to time (T+$\Delta$T/2) and flagellar length (L +$\Delta$L/2). To record the flagellar growth with high-temporal resolution, $\Delta$T needs to be short. However, it cannot be too short because flagellar growth $\Delta$L within $\Delta$T has to be resolvable by the optical resolution of our imaging system, which is one pixel corresponding to 65 nm physical distance.

## Flagella grow at a constant rate when their length is short

Firstly, we aim to investigate the flagellar growth rate at different flagellar length by performing time-lapse imaging of a large population of cells. According to a previous study (*Iino, 1974*) and our observation (*Figure 1—figure supplement 2*), the average flagellar length of a bacterial population increases with the culturing time. After preliminary experimental tests, *V. alginolyticus* after 1.5 hr culturing in VPG broth (Materials and methods), representing cells with short flagella, were imaged to investigate the flagellar growth rate. A time interval of 5 min was adopted in the time-lapse imaging since flagellar growth rate in this regime is relatively fast. We collected images from 72 cells with flagellar lengths ranging from 400 nm to 2800 nm and the instantaneous flagellar growth rates vs. flagellar length were plotted in blue dots in *Figure 2c*. Each data point is from one individual cell and its growing flagellum. The growth rate is nearly constant when the flagellar length is shorter than ~1500 nm and then drops sharply. The initial growth rates are between 37 and 60 nm/min for the flagella shorter than 1500 nm. Within the 5-min measurement interval, the flagella can grow 185–300 nm. Ultra-short flagella (<400 nm) are not able to attach to the surface and therefore accurate measurement is difficult.

## Flagellar growth rate decays sharply when the length is long

To investigate the growth rate of longer flagella, *V. alginolyticus* cells after 2.5 hr culturing in VPG broth (Materials and methods) were imaged. Since the flagellar growth rate starts to drop when filaments are longer than ~1500 nm, a 20-min time interval was adopted in the time-lapse imaging. We

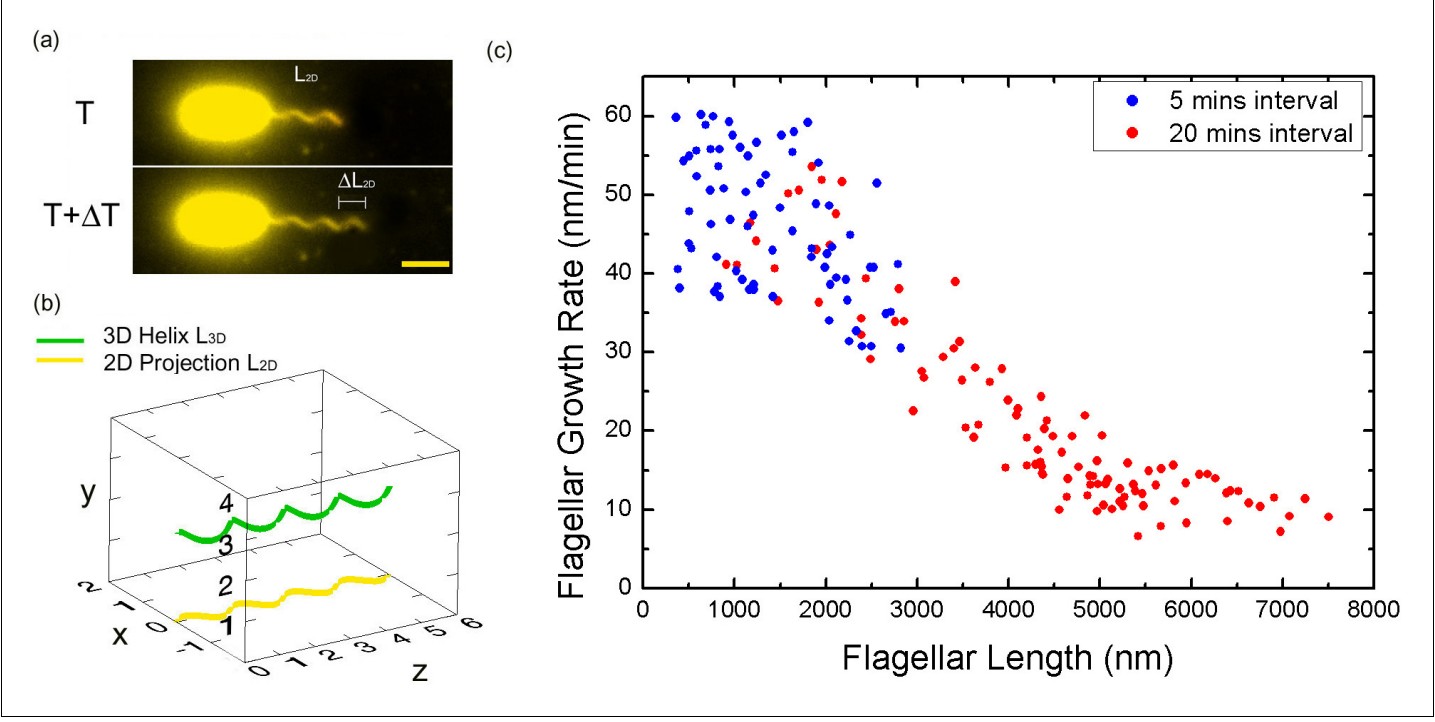

**Figure 2.** Bacterial flagellar growth rate measurements. (**a**) Representative images of a growing bacterial flagellum, taken at different time points, $\Delta T$ = 30 mins. The flagellar length increases by length $\Delta L_{2D}$. The growth rate of the flagellum can be calculated as $\Delta L/\Delta T$. Scale bar: 2 μm. (**b**) Schematic of a helical flagellum in 3D space $L_{3D}$ (green) and its 2D projection $L_{2D}$ (yellow). (**c**) Length dependence of flagellar growth rate of *Vibrio alginolyticus* from measuring growth rate at different flagellar lengths at 5 min intervals for short flagella (blue points) and 20 min intervals for longer flagella (red points). Flagellar growth rate is constant for short flagella (<1500 nm), and it decreases dramatically with increasing flagellar length.

The following figure supplements are available for figure 2:

**Figure supplement 1.** Measurement of flagellar length.

**Figure supplement 2.** Flagellar growth rate measured from non-tethering flagella.

collected images from 103 cells with flagellar lengths ranging from 1000 to 7500 nm. The instantaneous flagellar growth rates vs. flagellar length were then plotted in red dots in *Figure 2c*. The growth rate decreases rapidly as the flagellar length increases from ~1500 nm to ~7500 nm. Overall, the growth rate starts in the range of ~50 nm/min and drops sharply afterwards and decreases to ~9 nm/min at 7500 nm. Our flagellar growth rate measurement is not sensitive to the chosen time interval since there is a region (between 1000 and 3000 nm) where both our 5 min and 20 min data overlap and agree. This provides assurance that the observed decay in flagellar growth rate is not an artifact caused by different measuring intervals and culturing time.

## Flagellar growth rate when it is free to move

To further prove the observed length dependence of flagellar growth rate is not an artifact caused by attaching flagella to the surface, we conducted a similar measurement of *V. alginolyticus* flagellar growth rate when the flagella are free to move in space.

In this experiment, the observation was made when bacterial cell bodies were immobilized on the surface and before the flagella bound to the surface. The fluorescent images of flagella are taken using confocal microscopy (Materials and methods) to catch its trajectory in 3D space. In most cases, although fluorescent flagella are discernible, their fine helical structure is blurred due to their rapid movement. We can only analyze very limited cells in which accurate flagellar length measurement is possible. *Figure 2—figure supplement 2a* displays some representative examples. Using the same definition we introduced earlier, the instantaneous flagellar growth rate can be calculated (*Figure 2—*

*figure supplement 2b*, black dots). We see that although the data points are not many, the sharp decay in flagellar growth rate as the flagellar length increases has been reproduced, confirming our observation of tethered cells.

## Continuous time-lapse recording of single *V. alginolyticus* polar flagellum growth

With particular interest, we applied our method to monitor the flagellar growth of single cell in real time. In this experiment, *V. alginolyticus* cells were imaged every 5 min. Owing to the high temporal and spatial resolution offered by our experiment, we were able to follow the in vivo flagellar growth dynamically. *Figure 3a* shows representative continuous time-lapse recording of a single bacterial cell and its growing flagellum. In *Figure 3b*, the corresponding flagellar length vs. time curve is presented as the red line. The instantaneous flagellar growth rate is then calculated as the length difference between two adjacent images divided by 5 min, as shown in *Figure 3c*, on top of the population measurement data from *Figure 2c*. In contrast to previous reports of a constant growth rate in *E. coli* or a monotonically decreasing rate in *S. typhimurium* (*Iino, 1974*; *Aizawa and Kubori, 1998*; *Turner et al., 2012*), we found that in *V. alginolyticus* the flagellar length increases linearly with time when its length is short (*Figure 3b*), which indicates a constant growth rate. When the flagellum grows longer than 1500 nm, the relationship between flagellar length and time deviates from linearity (*Figure 3b*, the red line), showing a decrease in growth rate. Therefore, we reproduced in single cells the length-dependent flagellar growth seen in population studies, although we were not able to follow continuous growth of flagella longer beyond 4000 nm due to the photobleaching of NanoOrange. To our knowledge, this is the first comprehensive report of in vivo real-time recording of *V. alginolyticus* single flagellar growth.

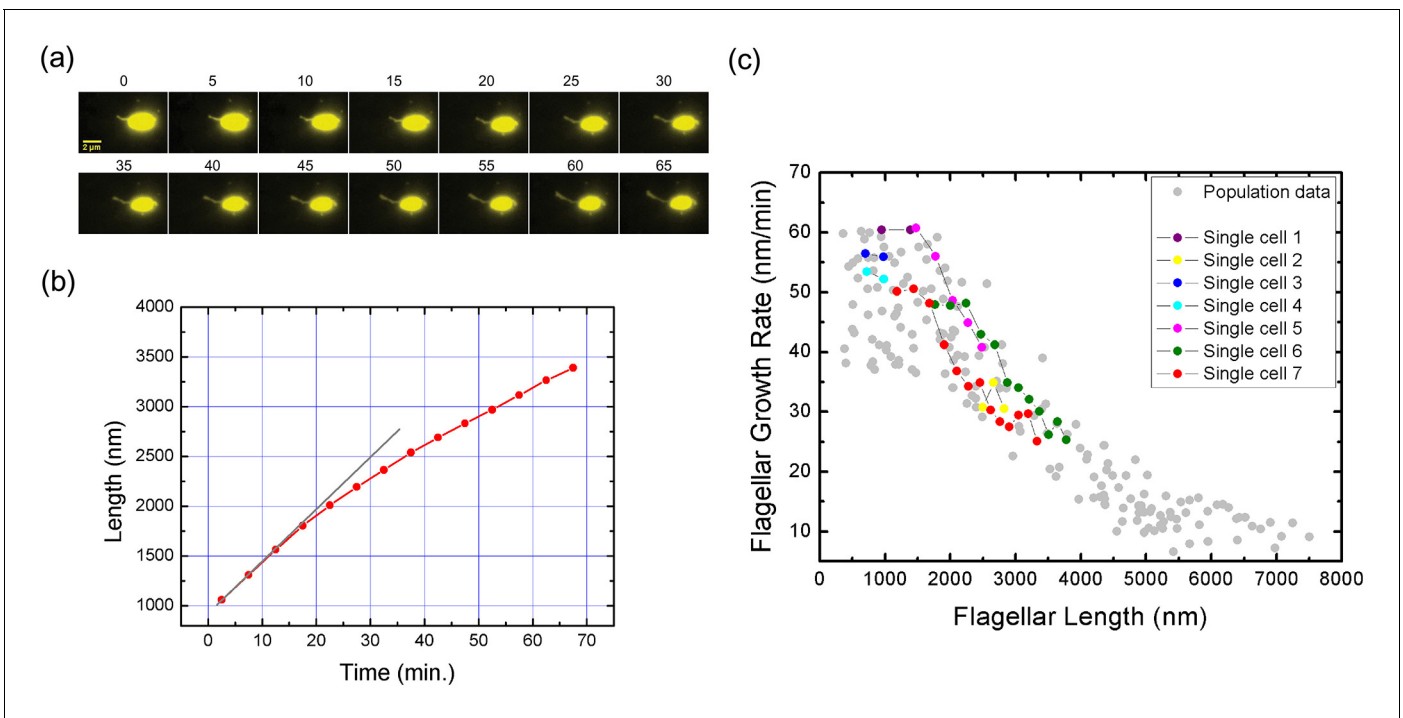

**Figure 3.** Real-time tracking of single flagellar growth by fluorescence microscopy. (a) Continuous time-lapse recording of a single growing flagellum with images taken every 5 mins. (b) Flagellar length versus time for the cell presented in (a) (red dots). Gray line shows initial slope of flagellar length versus time. (c) Population measurement data for flagellar growth (grey dots) with measurements from cells tracked over more than two frames indicated in color. The cell in (a) and (b) is indicated in red.

## Simulating flagellar growth by an injection-diffusion model

To explain why the growth rate of flagellar filaments undergoes two phases, we present a mathematical model incorporating current knowledge about the system. We construct our model as follows:

1. At the base of the BFM, flagellin monomers are inserted into the channel by the T3FSS at a constant loading rate S$_{load}$, implementing free energy derived from ATP hydrolysis and PMF (*Paul et al., 2008*; *Minamino and Namba, 2008*; *Lee and Rietsch, 2015*).
   The channel is so narrow (*Yonekura et al., 2003*) that only in its partially unfolded form as a peptide chain can flagellin travel through. *V. alginolyticus*'s unfolded flagellin monomer is modeled as a cylinder like peptide chain of ~2 nm in diameter by 56 nm long (Materials and methods). To simulate the secretion force applied by the secretion apparatus, we introduce a free parameter of the model, loading strength (*LS*), assuming pushing from the apparatus can enforce the movement of flagellin monomers in the channel within the distance of $LS \times 56$ nm. In the special case of $LS = 0$, it will wait until the first 56 nm space of flagellum is empty and then flagellin enters. If there is already a flagellin waiting for entering, the new-coming flagellin will diffuse away (*Figure 4a*). When $LS = 1$, the entering flagellin can only push one flagellin in the channel to create up to 56 nm in space for its entering. If two flagellins are needed to be pushed to make enough space, then a new flagellin would fail to enter the channel.

2. Once pumped into the channel by the secretion apparatus, the flagellin cannot move back to travel through the base and return to the cytoplasm. Partially unfolded flagellin monomers, ~2 nm in diameter by 56 nm long (Materials and methods), travel in the channel as a single-file diffusion (*Figure 4a*). The movement of each flagellin in the channel can be described by the Langevin equation:

$$m\frac{d^2x}{dt^2} = F - \zeta\frac{dx}{dt} + \eta(t) \tag{1}$$

where $m$ denotes the mass of the flagellin monomer, $x$ its position in the channel, $F$ the external force driving the motion of each monomer, $\zeta$ the viscous drag coefficient, and η(t) the Brownian force due to thermal fluctuations, which satisfies the white-noise property.

$$\langle \eta(t) \rangle = 0 \tag{2}$$
$$\text{cov}[\eta(t)\eta(s)] == \langle \eta(t)\eta(s) \rangle = 2kT\zeta\delta(t-s) \tag{3}$$

Since the environment in which bacteria live is one of low Reynold's number, inertial forces can be neglected and removed from the Langevin equation (*Purcell, 1977*). Previous research also indicates that when flagellin monomers are traveling in the channel, no external force is applied to drive its movement (*Yonekura et al., 2003*; *Stern and Berg, 2013*). Therefore, *Equation 1* can be simplified as

$$\zeta\frac{dx}{dt} = \eta(t) \tag{4}$$

By introducing a Wiener process, *Equation 4* can be numerically simulated following:

$$x(t+\Delta t) = x(t) + \sqrt{2D\Delta t} \cdot Z \tag{5}$$

where $Z \sim N(0,1)$, $D$ the apparent diffusion coefficient of flagellin monomers in the channel, and $\Delta t$ the time interval of one simulation step.

3. The channel is so narrow that one flagellin cannot move beyond another. When one motion leads to collision between two flagellins, this motion pauses at a position where the adjacent flagellins meet. The same is true for all the flagellin subunits in the channel: if other flagellins are in the way, they stop moving when touching their neighbors at that time step.

4. At last, when a flagellin reaches the tip of the growing flagellum, it crystallizes immediately and leaves the simulation. At this moment, this flagellin becomes the new extension of the flagellum and the flagellum elongates for $\Delta L$, as shown in *Figure 4a*. As one turn of 11 flagellins elongates the flagellum by 5.2 nm (*Yonekura et al., 2003*), $\Delta L$ can be estimated as 5.2 nm/ 11 = 0.47 nm.

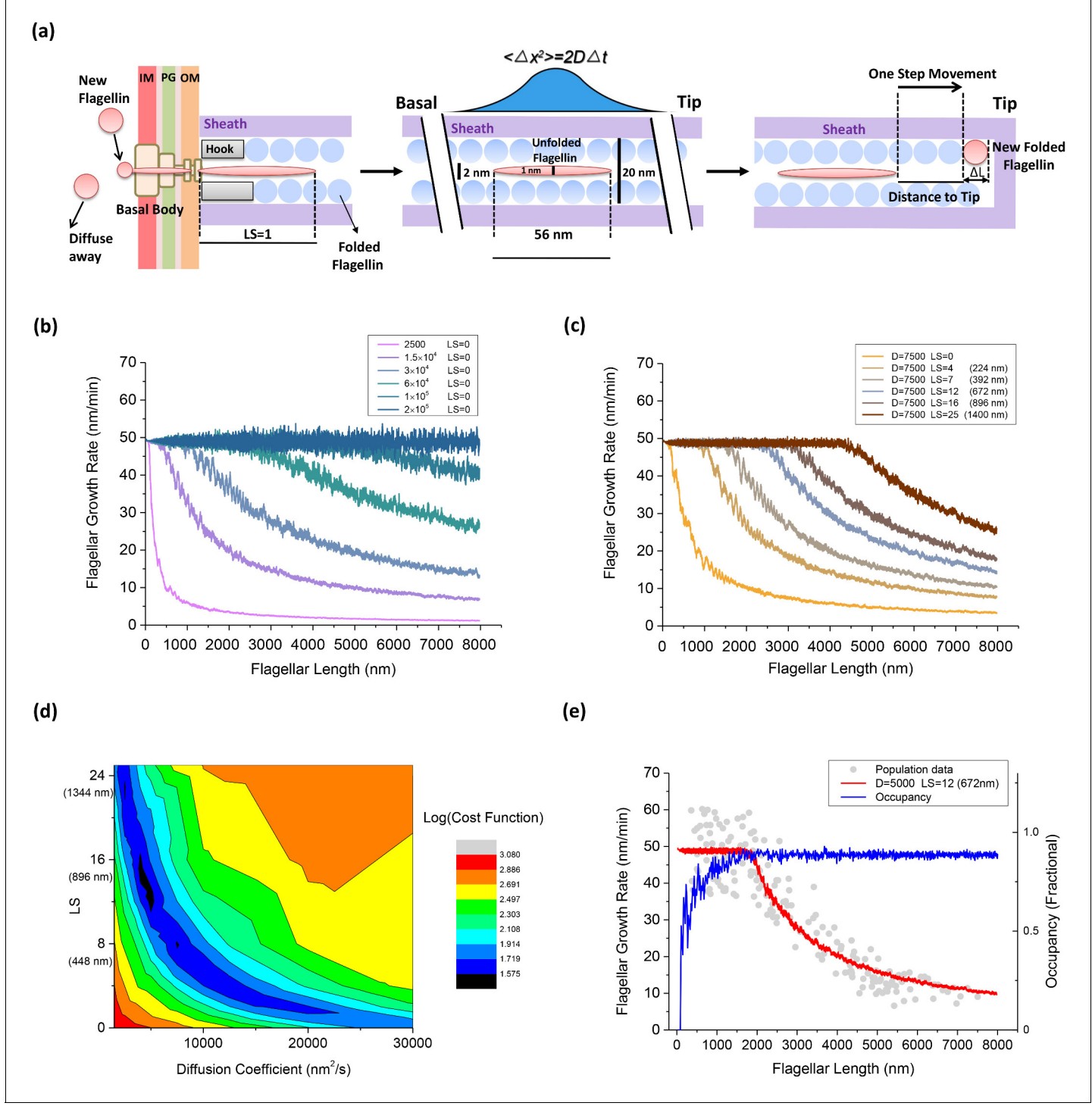

**Figure 4.** An injection-diffusion model explaining the length dependent flagellar growth. (**a**) Schematic illustration of the key elements of our one-dimensional diffusion model. At the base of the BFM, flagellin monomers are inserted into the channel by the T3FSS at a constant speed $S_{load}$. Once pumped into the channel, the flagellin diffuses as an unfolded peptide chain of 56 nm long. When a flagellin monomer reaches the tip of the growing flagellum, it crystallizes immediately and extends the flagellum by 0.47 nm. (**b**) Simulated flagellar growth rate vs. length curves with loading strength equal to zero (LS = 0) and varying values of D. (**c**) The simulated flagellar growth rate vs. length curves with D = 7500 nm$^2$/s and varying values of LS. (**d**) The contour plot representing the cost function of the model fitting to experimental dataset in the parameter space formed by D and LS. (**e**) The predicted flagellar growth rate vs. length curve of the best-fit parameter set (LS = 12 and D = 5000 nm$^2$/s, red line) and the channel occupancy vs. length (blue line).

To simulate flagellar growth, we implemented the above process by Monte Carlo simulations. In each simulation step, every flagellin monomer in the channel is polled according to equation (*Equation 5*) to update its location in the channel. At the same time, the new location of each monomer is checked for (1) if it contacts its neighbors, the movement stops; (2) if it has reached the growing tip, it becomes the new extension of the flagellum and leaves the simulation; (3) if it returns to the secretion apparatus, the movement stops at the base.

Using this program, movement of each flagellin molecule can be tracked and flagellar growth with high temporal resolution can be simulated. Following the experimental procedure for calculating the instantaneous flagellar growth rate, the model can predict the relationship between flagellar growth rate and flagellar length. There are three critical parameters of this model: the loading rate $S_{load}$, the loading strength $LS$ and the diffusion coefficient $D$ of the flagellin monomer in the channel. According to our experimental data (*Figure 2c*), when flagella are short, we observed a plateau in growth rate, which is also the maximum possible growth rate of the system. By extrapolation, this maximum growth rate is estimated as 50 nm/min that corresponds to transporting 1.7 flagellins per second. The maximum growth rate also reveals the number of working cycles of the T3FSS. Therefore, $S_{load}$ was experimentally determined as 1.7 Hz. To fit the experimental curve, we varied $D$ and $LS$, two free parameters of the model, to optimize the fitting.

## Model parameter space search and the best fit

We varied $D$ from 1500 to $2.0 \times 10^5$ nm$^2$/s, $LS$ from 0 to 25, to evaluate the fitting between our model prediction and experimental result. In our model, we first set $LS = 0$ and saw the effect of varying $D$. When $D$ was greater than $2.0 \times 10^5$ nm$^2$/s, every flagellin quickly diffuses to the tip once entering the channel. In this scenario, the flagellar growth rate is determined solely by the loading speed $S_{load}$, and therefore, it was a constant rate spanning across a large range of flagellar lengths, as shown in *Figure 4b* (blue line). With decreasing $D$ values, the plateau in growth rate vanishes gradually and the growth rate decays sharply due to increasing futile movement of flagellins in the channel. When $LS = 0$ and $D$ was less than 2500 nm$^2$/s, as shown in *Figure 4b* (magenta line), the flagellar growth rate dropped directly to ~zero at very short lengths. In this condition, the flagellin transportation is inefficient and since T3FSS does not enforce flagellin movement and the flagellum can hardly grow.

We also tested the effect of varying $LS$. According to previous research, the T3FSS at the base of the flagellum implements free energy derived from ATP hydrolysis and PMF to insert flagellins into the channel, although the exact magnitude of the pushing force has not been resolved. In *Figure 4c*, we set D = 7500 nm$^2$/s and varied $LS$ from 0 to 25. When $LS = 0$, as shown in *Figure 4c* (orange line), the flagellar growth rate drops directly to ~zero at very short lengths, similar to the magenta line in *Figure 4b*. With increasing $LS$, the effect is immediately shown as a plateau emerging in the flagellar growth rate in the short flagella regime (*Figure 4c*) and the width of the plateau extends as the $LS$ increases.

The best fit of model prediction to experimental data is found by minimizing the cost function (Materials and methods). A two-dimensional parameter space search was conducted and the corresponding value of the cost function is plotted in contour plot in *Figure 4d*. The best fit was found at $LS = 12$, D = 5000 nm$^2$/s. In *Figure 4e*, we overlaid the predicted growth rate from the model with the population measurement dataset. Also, we calculated the channel occupancy defined as the total length of flagellins in the channel divided by the entire length of the flagellum. In the short flagella regime, the constant flagellar growth rate is supported by the secretion force of the T3FSS and fast diffusion of flagellins across short distances. At flagella lengths larger than 1500 nm, the rate-limiting effect of flagellins diffusing across a large distance became increasingly prominent, causing many flagellins to jam in the channel (hence the high channel occupancy) and the growth rate dropped rapidly.

## Discussions

In this work, we reported a fast and high-resolution fluorescent labeling method that enables in vivo real-time recording of *V. alginolyticus* polar flagellar growth. We found that the flagellar growth rate of *V. alginolyticus* is length-dependent, showing distinct growth regimes. Initially, the flagellum

grows at a constant rate when its length is below 1500 nm. While it continues to grow, the growth rate decreases sharply.

We successfully executed real-time recording of flagellar growth by labelling the flagellar sheath of *V. alginolyticus*. Although we have experimental evidence to prove the synchronized growth between sheath and flagella, the exact mechanism driving sheath growth remains a mystery. From the TEM images, we see that sheath forms very compact and organized structure, always wrapping the inner flagellum. This raises another interesting question: how does the sheath self-assemble? It may be that sheath components are secreted through the central channel of the filament following a similar mechanism as the flagellar assembly. The sheath may also have been formed along the exterior of the flagellum, driven by the high motility of lipids. The exact mechanism of sheath assembly, and the relative kinetic comparisons to flagellar growth have not yet been elucidated, and we hope to pursue this in a subsequent study.

It is perplexing that flagellin transit is maintained in the channel without an explicit energy source (*Minamino et al., 2014*). Recently, to explain the constant flagellar growth rate of *E. coli*, two models have been proposed, the single-file diffusion model (*Stern and Berg, 2013*) and the chain model (*Evans et al., 2013*). First, the single-file diffusion model without invoking an injection pushing mechanism is a special case of our model when $LS = 0$. As demonstrated in *Figure 4b*, to produce a constant flagellar growth rate, the diffusion constant $D$ must be very large to make the loading speed the rate-limiting step. In comparison to our length-dependent flagellar growth data with both a plateau regime and a fast-decay regime, varying $D$ alone is not sufficient to fit the whole curve, as shown in *Figure 4d*. Therefore, a single-file diffusion model without invoking a push mechanism cannot explain our data. Secondly, the chain mechanism was proposed to explain the constant rate of flagellum growth. In this model, flagellin monomers are connected to each other through a head-to-tail linkage, forming a continuous chain in the channel and the entropic force generated from crystallization of flagellin monomer at the growing tip pulls the flagellin chain through the channel. Although this model offers a simple physical mechanism for the constant growth rate, it seems difficult to explain the sharp decay in growth rate of *V. alginolyticus* when its flagellum grows longer. Moreover, in the case where the multi-subunit chain breaks, the chain of distal subunits is pulled to the growing tip, leaving the chain of basal subunits to diffuse in the channel. Flagellum growth resumes only when the chain of basal subunits reaches the growing tip and crystallizes. During this period, the growth rate decays to zero. One direct prediction from this mechanism is that flagellar growth is not continuous and that pauses in flagellar growth will be observed at different flagellar lengths. In our experiments, we observed neither a constant growth rate nor pauses in flagellar growth. Therefore, our data is not explained by the chain model. Alternately, we have proposed here an injection-diffusion model with injection force that explains our experimental results.

One critical component of the injection-diffusion model is that we have to allow the export apparatus to be able to force the flagellin movement within $LS$ distance. Due to this effect, when flagella are short, they can move fast toward the growing tip even when $D$ is not extra large, leading to a constant growth rate up to 1500 nm. At this stage, the flagellar growth rate is limited by the flagellin loading rate from the secretion system. As this effect vanishes when flagella grow longer, the growth rate drops sharply, as a result of futile diffusion steps of flagellin monomers in the channel. Meanwhile, some flagellins begin to jam in the channel. Such blockage impedes the loading of new flagellins at the base, further reducing the growth rate. Finally, when the flagella are long enough, slow diffusion leads to a high occupancy of flagellins in the channel, whereupon the flagellar growth rate relies again on the secretion force of the export apparatus, but at a much decreased export frequency, therefore converging to a discounted loading speed.

To justify the secretion force in our model, we estimated the energy required to push flagellins in the channel. From the Einstein-Smoluchowski relation, the viscous drag coefficient of a single flagellin in the channel is $\zeta=kT/D$. The work required to push $n$ flagella $s$ distance at velocity $v$ is $W=n\zeta vs=nkTvs/D$. To estimate to maximum energy required for pushing, we consider the condition as follows. In the condition where $n = 12$ flagellins are close-packed occupied in the beginning of channel, the secretion machine must push the new-coming flagellin 56 nm forward into the channel, as well as the 12 flagellins ahead of it. Using the best-fit parameter set ($LS = 12$, $D = 5000$ nm$^2$/s), we can calculate the free energy required for this insertion. If we assume this insertion can be finished within 0.1 s, that gives v = 56 nm/0.1 s = 560 nm/s and W = 75.3kT = 3.77 ATP. If we assume this, insertion takes the entire cycling time of the secretion apparatus, which is 0.6 s, that gives

v = 56 nm/0.6s = 93 nm/s and W = 12.5kT = 0.62 ATP. Previous research suggests that the FliI ATPase at the core of the T3FSS shares similar molecular structure with the F1 ATPase (*Ibuki et al., 2011*) that can rotate the central shaft for one revolution by hydrolyzing 3 ATP molecules. Although the details of force generation and energy transduction in the T3FSS remain unresolved, from our estimation we see that the hypothesized secretion force applied by the apparatus is reasonable and consistent with our current knowledge about the system.

Our experimental work and theoretical analysis suggest that the physical principle of diffusion is implemented here for extracellular protein transport. We see that a simple one-dimensional diffusion with a pushing base and an absorbing tip can support flagellar growth and explain its length-dependent features. From the plateau in growth rate when the flagellum is short, we deduce the loading speed of 1.7 flagellin per second of the *V. alginolyticus* T3FSS. Compared to the results from Turner et al. (*Turner et al., 2012*; *Stern and Berg, 2013*), in which the loading of 0.5 flagellin per second was estimated in *E. coli*, our result is more than three times faster. This difference is possibly due to the difference between bacterial species. More experimental investigations are needed to characterize the length-dependence in flagellar growth in *E. coli* and *S. typhimurium*, to look for rate-change features. The previously reported constant growth rate in *E. coli* and exponential decay rate in *S. typhimurium* may be unique to that bacterial species, or may correspond to different regimes of a universal length-dependent growth curve, similar to the one we reported here. The details of the injection mechanism and diffusion process can be further elucidated through mutagenesis or energetic control of cells in the future.

## Material and methods

### Bacterial strains and culture conditions

The *V. alginolyticus* strain used in this study is VIO5, a lateral flagella deletion strain (gift from Dr. Michio Homma, Nagoya University, Japan (*Furuno et al., 1997*). The strain was grown aerobically with shaking from frozen stocks (made from single colonies) to saturation at 30°C in VC medium (0.5% Bacto-tryptone (BD) (w/v), 0.5% yeast extract (Sigma Aldrich) (w/v), 0.4% $K_2HPO_4$ (w/v), 3% NaCl (w/v), 0.2% glucose (w/v)) overnight, then transferred to VPG broth (1% Bacto-tryptone (BD) (w/v), 0.4% $K_2HPO_4$ (w/v), 3% NaCl (w/v), 0.5% glycerol (w/v)) for 1.5 hr to an $OD_{600}$ = 0.3. The bacterial culture was subsequently diluted 1:5 into the imaging medium containing VPG broth with 15% TMK medium (50 mM Tris-HCl (pH = 7.5), 5 mM $MgCl_2$, 5 mM Glucose, 300 mM KCl).

### Visualization of sheathed flagella by transmission electron microscopy

The experiment was performed as follows: the bacteria were grown overnight in VC medium at 30°C, diluted 1:100 in VPG broth and incubated for 3 hr at 30°C. For the NanoOrange group, Nano-Orange was added into VPG broth during culturing (1/100 v/v); For the fixation group, the cells were fixed by 1% formalin for 24 hr and washed with distilled water. A drop (20 to 35 μl) of cell culture was placed on a 200-mesh carbon Formvar-coated grid for 2 mins and the excess fluid was removed with a filter paper. Cells were then negatively stained for 1 min with 1% w/v uranyl acetate. The grids were viewed under a Tecnai G2 20 Twin transmission electron microscope (TEM) at an accelerating voltage of 120 kV.

### Sample preparation and flagella labeling

Microscope tunnel slides were constructed by attaching double-sided tapes between microscope slides and cover slips, which were cleaned in a saturated solution of KOH in 95% ethanol. Poly-L-lysine (Sigma, 0.1%) was added into the tunnel for 1 min to coat a positively charged layer on the glass surface, and then flushed out with the imaging medium. Labeling of the flagella was performed by mixing NanoOrange (Invitrogen, Carlsbad CA, 1/100 v/v) with the diluted VIO5 cells gently. The stained *V. alginolyticus* cells were then injected into the tunnel for 2 min for their immobilization. Unattached cells were flushed out with the imaging medium.

### Super-resolution imaging by SIM

To obtain high-quality images of the bacterial flagella, super-resolution fluorescent imaging was performed using the Structured Illumination Microscopy (Nikon N-SIM). The microscope is equipped

with an EMCCD camera (Andor iXon DU-897), a 100 × 1.49 NA TIRF objective (Nikon CFI Apo TIRF), a 488 nm excitation laser (Coherent Sapphire 488 LP), and a bandpass emission filter (500–550 nm, Chroma). Image acquisition and reconstruction were directed by the NIS-Elements Viewer (Nikon) software.

## Dynamic flagellar growth imaging

For the real-time observation of flagellar growth, the prepared cells were imaged under an automatic fluorescence microscope (Nikon, Ti-E) equipped with a 1.49 NA CFI Apo TIRF 100x Oil objective, a 470 nm LED light source (Omicron, LEDHub), a sCMOS camera (Andor, Zyla) and the perfect focus system. The tunnel containing stained cells was set in a humidity chamber with a temperature-control system to allow flagellar growth at 30°C. During our imaging, NanoOrange was kept in the imaging medium (1/100 v/v) to ensure timely labeling of newly grown flagellar segments. Because the sheathed flagella readily attach to the poly-L-lysine coated surface, both the cell body and its flagellum are well immobilized on the microscope tunnel slides, enabling several hours of observation to track the flagellar growth. Time-lapse fluorescent images were captured at 5/20 min intervals depending on the flagellar growth rate.

## Image analysis and flagellar growth rate measurement

The acquired images were analyzed using the ImageJ software. With our imaging system, the physical distance corresponding to one pixel is 65 nm. For the flagellum of interest, the ImageJ free-hand line tool was used to trace filament contours. Then the contour trajectories on the image were smoothed by applying the ImageJ fit spline function and the 2D length ($L_{2D}$) measurement of each flagellum can be obtained (**Turner et al., 2012**).

2D projection correction: The fluorescent images of flagella are the 2D projections of the 3D helical filaments with 0.16 µm helix radius and 1.5 µm pitch (**Furuno et al., 1997**), **Figure 2b.** The helix can be written as $x=rsin(2\pi T)$, $y=rcos(2\pi T)$, $z=cT$, where $r$ is the helix radius, $T$ the turns, $c$ the pitch. The 3D helix length can be expressed as

$$L_{3D} = \sqrt{\left(\frac{2\pi rz}{c}\right)^2 + z^2} \tag{6}$$

The 2D projection is then $x=rsin(2\pi T)$, $z=cT$ and can be rewritten as $x=r\,sin(2\pi z/c)$. The arc lengths of 2D projection can be calculated numerically by

$$L_{2D} = \int_0^Z \sqrt{1+(f')^2}dz = \int_0^Z \sqrt{1+\left(\frac{2\pi r}{c}cos\left(\frac{2\pi z}{c}\right)\right)^2}dz \tag{7}$$

Therefore, $L_{3D}$ and $L_{2D}$ can be calculated in respect to z and plotted in **Figure 2—figure supplement 1(c)**. The linear fit that we used as our correction is $L_{3D} = 1.09\,L_{2D}$.

In a long time tracking experiment, single flagellar growth can be recorded in the curve of flagellar length versus growth time (**Figure 3b**). Consequently, the flagellar growth rate can be calculated as the length difference between adjacent two images of the same filament divided by the measurement interval (5 or 20 min). This rate was assigned as the instantaneous growing speed of the flagellar length corresponding to the midpoint of the newly grown segment. In this way, the flagellar growth rate versus flagellar length can be constructed (**Figure 2c**, **Figure 3c**).

## Non-tethered flagella growth-rate measurement

To rule out the possibility that our observed length dependence in flagellar growth rate was an artifact caused by attaching flagella to the surface, we conducted a flagellar growth rate measurement when the *V. alginolyticus* flagella were free to move. Images of growing flagella swinging in space were captured by a high-speed confocal microscopy (Olympus, IX71 and Fluoview FV300) to measure the length of non-tethered flagella. **Figure 2—figure supplement 2a** showed pairwise confocal images of the same flagella recorded at 10 mins time interval. The growth rate was calculated following the strategy we introduced in the main text, and the data points were plotted in **Figure 2—**

*figure supplement 2b* (black dots), which indicates a similar trend to the flagellar growth rate measured using tethered cells (gray dots).

## Estimation of *V. alginolyticus* flagellin length

The *V. alginolyticus* flagellin protein FlaE (Uniprot ID D0X0U4) consists of 379 amino acids, which is shorter than the *Salmonella enterica* flagellin protein FliC of 494 amino acids. The structure of folded FlaE was predicted by homology modeling Swiss-Model server using *S. enterica* FliC (1UCU) as the template. The predicted structure is similar to FliC in the flagellar channel core region (D0 and D1) but less dense on the outer region (D2 and D3), due to its shorter sequence. Hence, the height and width of folded FlaE is the same as FliC. When a flagellin reaches the tip of the growing flagellum, it crystallizes immediately and this flagellin becomes the new extension of the flagellum. As one turn of 11 flagellins elongates the flagellum by 52 Å in *Salmonella enterica* (*Yonekura et al., 2003*), the growth increment of each folded FliC can be estimated as 52 Å/11 = 0.47 nm. Therefore, $\Delta$L of our model (*Figure 4*), the growth increment of each folded FlaE can be estimated as 0.47 nm.

When the flagellin is transported in the channel, it has to pass through in the form of a peptide chain. We then assumed the length of the unfolded peptide chain of a protein is linear to its sequence length of amino acids. Given the length of the partially unfolded FliC is 74 nm (*Yonekura et al., 2003*; *Turner et al., 2012*) conserved alpha helix, the length of the unfolded FlaE can be estimated as $74 \times 379 \div 494 = 56$ nm.

## Generating flagellar growth sample traces

Following the model framework introduced in the main-text, a custom computer program written in C++ was implemented to generate flagellar growth sample traces. Each sample trace is simulated for the length of up to 8 μm. To balance numerical accuracy and computational load, the simulation time step $\Delta$t was chosen as $(\Delta L)^2/D$ for different diffusion coefficient of the flagellin monomer. We tested simulations with smaller $\Delta$t and confirmed the numerical convergence of simulated results.

## Calculating simulated flagellar growth rates

Once we obtained the simulated flagellar length vs. time traces, we calculated the instantaneous flagellar growth rate following the experimental procedure: at each time point, the instantaneous flagellar growth rate was defined as the length difference between the current time and 5 min later, divided by 5 min. This growth rate was assigned to the current length plus half of the length difference. After this process, we obtained a preliminary growth rate versus length relationship. This curve was filtered by a median filter of 100 data points and then, the smooth growth rate versus length trace was generated for presentation and subsequent calculation of the cost function.

## Cost-function evaluation

For a comparison between the experimental dataset and the predicted growth rate vs. length relationship, a cost function was constructed and used to evaluate the goodness of fitting. For every experimental data of the in total N data points, its length value ($L_i$) was retrieved and put into the model predicted curve to find its corresponding growth rate ($V_i model$). The cost function between a given model predicted curve and the experimental dataset is defined as

$$E(D, LS) = \sum \left[ (V(L_i) \, _i model - V(L_i) \, _i exp)^2 \right]/(n-1).$$

## Contour plot formation

After generating different flagellar length vs. time traces of different combinations of diffusion coefficients (*D*) and loading strength (*LS*), we calculated their corresponding cost function in fitting to the experimental dataset. The values of these cost functions have been plotted in contour plot on the *D-LS* plane and then we can search for the best fit with the minimum value of the cost function.

## Acknowledgements

This work is financially supported by the Ministry of Science and Technology, Republic of China, under contract No. MOST-103–2112 M-008-013-MY3 to CL, the National Natural Science Foundation of China (No. 31370847, No.31327901), and the Recruitment Program of Global Youth Experts

to FB. CL and FB are also supported by the Human Frontier Science Program Grant (RGP0041/ 2015). We also thank Dr. Wei-Chang Lo for the valuable discussion and Chunyan Shan, Yingchun Hu and Yunchao Xie in the Core Facilities of Life Sciences, Peking University for assistance with SIM and TEM imaging.

## Additional information

### Funding

| Funder | Grant reference number | Author |
| --- | --- | --- |
| National Natural Science Foundation of China | No. 31370847 | Fan Bai |
| Human Frontier Science Program | RGP0041/2015 | Fan Bai Chien-Jung Lo |
| National Natural Science Foundation of China | No.31327901 | Fan Bai |
| Ministry of Science and Technology, Taiwan | MOST-103-2112-M-008-013-MY3 | Chien-Jung Lo |

The funders had no role in study design, data collection and interpretation, or the decision to submit the work for publication.

### Author contributions

MC, Conceptualization, Data curation, Formal analysis, Writing—original draft; ZZ, Data curation, Formal analysis, Writing—original draft; JY, Data curation, Software, Formal analysis, Writing—original draft; KP, Data curation, Formal analysis; MABB, Formal analysis, Writing—original draft, Writing—review and editing; FB, Conceptualization, Formal analysis, Supervision, Funding acquisition, Investigation, Methodology, Writing—original draft, Project administration, Writing—review and editing; C-JL, Conceptualization, Formal analysis, Funding acquisition, Investigation, Methodology, Writing—original draft, Project administration, Writing—review and editing

### Author ORCIDs

Chien-Jung Lo, http://orcid.org/0000-0002-8078-4970

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
