## [Decision Letter]

Thank you for submitting your article "Length-dependent flagellar growth of *Vibrio alginolyticus* revealed by real time fluorescent imaging" for consideration by *eLife*. Your article has been reviewed by four peer reviewers, one of whom, Richard M. Berry, is a member of our Board of Reviewing Editors and the process has been overseen by Richard Losick as the Senior Editor.

The reviewers have discussed the reviews with one another and the Reviewing Editor has drafted this decision to help you prepare a revised submission.

The authors have measured the rate of growth of the polar flagellum and sheath of *Vibrio alginolyticus*, using a vital dye, NanoOrange, whose fluorescence is low until it binds to the flagellar sheath (to a non-polar substrate). Thus, they can monitor changes in length without manipulating cells (e.g., washing and staining). This is a notable improvement. They show that the sheath is closely opposed to the filament. Presumably the sheath grows at its base where it is contiguous with the cell's outer membrane, but this is not known.

The experimental method and results add substantially to our understanding of flagellar filament growth, making the paper suitable in principle for publication in *eLife*. However, substantial improvements are needed in the data fitting and modelling, and in the discussion of how the results and model relate to existing models of flagellar-type export.

Substantial comments:

1) Data fitting: The 3-part piecewise linear fit to the speed vs. length data (Figure 2) is not very convincing. In fact, Figure 4 (Discussion, first paragraph) shows a different fit to the data than 3 piecewise linear segments. More care is needed to discuss what can and cannot be determined by fitting the data and comparing it to the results of modelling.

2) Further to comment 1): the model appears to be over-specified. It needs to be demonstrated that (or discussed whether) the data can equally well be explained by single-file diffusion (Stern and Berg, 2013) or a chain model (Evans et al., 2013) without invoking the pushing mechanism. It is not realistic to assume that a flagellin subunit that has diffused away from the base of the filament is not allowed to diffuse back. Nor is it necessary to assume that the subunits are completely unfolded, since an α-helical chain is only about 1 nm in diameter, and therefore smaller than the 2nm pore.

Preferably the modelling should be improved substantially. At least the discussion of other models in the literature (particularly Evans et al., 2013 and Stern and Berg, 2013) should be much improved and the need for the new model features (driven by data), and their plausibility, discussed carefully and quantitatively.

Further details on this point:

Discussion, third paragraph: All that is needed to explain these data is that there is a length dependence of the transit rate, which becomes rate-limiting (= slower than injection) as filaments grow longer. The length dependence of the chain model (Evans et al., 2013) is not discussed, so nothing can be said for or against it.

Subsection “Simulating flagellar growth by a one-dimensional single-file diffusion model”, point 1: "SI appendix" – I could not find. Presumably this justifies the use of 56 nm as the length of the flagellin in the channel. Discuss this in the text. In particular, how can you have LS >1 (best fit is LS=12)? This requires the apparatus to push filaments further than their length! If there is an extended length and a semi-coiled length, this needs to be said. Subsection “Estimation of *V. alginolyticus* flagellin length”, last paragraph: a simple estimate of the contour length based on the dimensions of a peptide bond (~0.35 nm per a.a.) gives about twice this contour length. Is 56 nm really a good estimate? How does the model and fit depend upon this?

Subsection “Simulating flagellar growth by a one-dimensional single-file diffusion model”, point 1: LS as described is really equivalent to a loading length in units of 56 nm – better to quote this length directly so that it can easily be compared to the filament length in Figure 4. Naively I'd expect that the occupancy saturates and the rate slows when the filament is longer than the loading length. But it seems that this happens at filament lengths *longer* than the loading length. Discuss/explain.

Subsection “Simulating flagellar growth by a one-dimensional single-file diffusion model”, point 3: Is it realistic that (both?) touching units stop? Rather than diffusing more slowly as a larger object?

Subsection “Model parameter space search and the best fit”, last sentence: Figures do not show that there is a dependence of growth rate in the limit of long filaments upon "secretion force" (LS?). If this has been demonstrated, it should be shown. It's not clear that the model is expected to predict this, especially in the light of the previous comment which seems to preclude monomers pushing each other. But perhaps in fact stopping upon contact DOES provide a mechanism for pushing, as hitting injected monomers will ratchet the Brownian Motion of those further in.

3) Subsection “Fluorescent labeling of flagellar sheath for real time monitoring of flagellar growth”, third paragraph: discuss/explain/fix the conflict between statements that flagella are anchored to the surface and the assumption that they are 3D helices. Are they helices anchored at lowest points? Is there any evidence for this, e.g. 3D imaging? The 2D-3D correction is minor to the point of being unnecessary, but also it is wrong if filaments are fully anchored to the surface and thus in fact 2D.

---

## [Author Response]

*Substantial comments:*

*1) Data fitting: The 3-part piecewise linear fit to the speed vs. length data (Figure 2) is not very convincing. In fact, Figure 4 (Discussion, first paragraph) shows a different fit to the data than 3 piecewise linear segments. More care is needed to discuss what can and cannot be determined by fitting the data and comparing it to the results of modelling.*

We thank the reviewers for pointing this out. We admit that our 3-part piecewise linear fit to the experimental data causes confusion. Therefore, we removed the 3-part linear fitting from the experimental data in Figure 2. The discussions regarding the third section of the 3-part linear fitting were subsequently removed from the text. Only the model simulation is used to fit the experimental data.

*2) Further to comment 1): the model appears to be over-specified. It needs to be demonstrated that (or discussed whether) the data can equally well be explained by single-file diffusion (Stern and Berg, 2013) or a chain model (Evans et al., 2013) without invoking the pushing mechanism. It is not realistic to assume that a flagellin subunit that has diffused away from the base of the filament is not allowed to diffuse back. Nor is it necessary to assume that the subunits are completely unfolded, since an α-helical chain is only about 1 nm in diameter, and therefore smaller than the 2nm pore.*

A) We followed the reviewer’s suggestion to discuss in more detail the single-file diffusion model and the chain model (also added to “Discussion”). First, the single-file diffusion model without invoking the injection pushing mechanism is a special case of our model when *LS*=0. As demonstrated in Figure 4), for *D*<3x10^4^ nm^2^/s, it is hard to reproduce the plateau phase in the growth rate when flagellum is shorter than 1500 nm. To achieve the plateau shown in our experimental data, it requires *D*≈4x10^4^ nm^2^/s, a value which seems unrealistic for flagellin diffusion in the channel. Additionally, a large *D* would result in slow decay in the growth rate when flagellum grows longer, which conflicts with our experimental observation. Therefore, we conclude that the single-file diffusion model without an additional injection pushing mechanism does not explain our data. Secondly, the chain mechanism was proposed to explain the constant rate of flagellum growth, in which the entropic force derived from flagellin crystallization automatically adjusts with the length of the multi-subunit chain to support a constant rate of subunit transit that is independent of channel length. However, in the case where the multi-subunit chain breaks, the chain of distal subunits is pulled to the growing tip, leaving the chain of basal subunits to diffuse in the channel. Flagellum growth resumes only when the chain of basal subunits reaches the growing tip and crystallizes. During this period, the growth rate decays to zero. One direct prediction from this mechanism is that flagellar growth is not continuous and that pauses in flagellar growth will be observed at different flagellar lengths. In our experiments we observed neither a constant growth rate nor pauses in flagellar growth. Therefore, our data is not explained by the chain model. Extra detail has been added to the comparison and discussion of these models in the third paragraph of the Discussion.

B) We believe there is some misunderstanding here. In subsection “Simulating flagellar growth by an injection-diffusion model”, point 2 it describes as “the flagellin cannot move back to the base”, which means the flagellin can diffuse forward or backward following the Langevin equation, but it is not allowed to diffuse back to leave the channel and return to the cytoplasm. We added further description in the aforementioned subsection of the revised manuscript.

3) We thank the reviewer for raising this question. We are sorry for being unclear in the manuscript. In fact, consistent with the reviewer’s suggestion, we assumed that the subunits in the channel are partially unfolded mainly as α-helices (Stern and Berg, 2013) and have further clarified this in our revised manuscript (subsection “Estimation of *V. alginolyticus* flagellin length”, last paragraph).

*Preferably the modelling should be improved substantially. At least the discussion of other models in the literature (particularly Evans et al., 2013 and Stern and Berg, 2013) should be much improved and the need for the new model features (driven by data), and their plausibility, discussed carefully and quantitatively.*

We followed the reviewers’ suggestions and substantially improved our modelling.

A) More details have been added to explain the new features of our model, such as a thorough introduction of Loading Strength (*LS*) in subsection “Simulating flagellar growth by an injection-diffusion model”, point 1.

B) Further discussion of other models such as the single-file diffusion model and the chain model has been included in the “Discussion” section (third paragraph).

C) Through comparison with other models, we have emphasized the plausibility of our model for our experimental data in the “Discussion” (third paragraph).

*Further details on this point:*

*Discussion, third paragraph: All that is needed to explain these data is that there is a length dependence of the transit rate, which becomes rate-limiting (= slower than injection) as filaments grow longer. The length dependence of the chain model (Evans et al., 2013) is not discussed, so nothing can be said for or against it.*

As we have mentioned above, the chain mechanism was originally proposed to explain a constant rate of flagellum growth. In this mechanism, the entropic force derived from flagellin crystallization adjusts as the multi-subunit chain grows in length to support a constant rate of subunit transit that is independent of channel length. However, in the case where the multi-subunit chain breaks, the distal subunits chain is pulled to the growing tip, leaving the basal subunits chain diffusing in the channel. The flagellum growth can resume only when the basal subunits chain reaches the growing tip and crystallizes. During this period, the growth rate decays to zero. A direct prediction from this mechanism is that flagellar growth is not continuous and paused flagellar growth would be observed at different flagellar lengths. We assume that longer the flagellum is, the more easily it breaks, at more locations. This reduces the growth rate of longer flagellum, which is caused by many long pausing events in subunit transit. However, such long pausing events in flagellar growth are not observed in our experiment. Therefore, our data are not well explained by the chain model.

*Subsection “Simulating flagellar growth by a one-dimensional single-file diffusion model”, point 1: "SI appendix" – I could not find. Presumably this justifies the use of 56 nm as the length of the flagellin in the channel. Discuss this in the text. In particular, how can you have LS >1 (best fit is LS=12)? This requires the apparatus to push filaments further than their length! If there is an extended length and a semi-coiled length, this needs to be said. Subsection “Estimation of V. alginolyticus flagellin length”, last paragraph: a simple estimate of the contour length based on the dimensions of a peptide bond (~0.35 nm per a.a.) gives about twice this contour length. Is 56 nm really a good estimate? How does the model and fit depend upon this?*

A) We are sorry: “SI appendix” is a typo. It has been replaced by “Material and methods”. The use of 56 nm as the length of the flagellin is stated in “Estimation of *V. alginolyticus* flagellin length” section in Material and methods.

B) The parameter loading strength (*LS*) assumes that pushing from the apparatus can enforce the movement of 56 nm of the new-coming flagellin monomer and any flagellin monomers ahead in the channel of it within distance of *LS*×56 nm, whose positions are affected by the insertion of the new-coming flagellin. It means that, when *LS*=0, the new flagellin will wait until the first 56 nm space of the channel is empty and then flagellin enters. When *LS*=1, the newly-inserted flagellin can only push one flagellin in the channel up to 56 nm in order to enter. If two flagellins are needed to be pushed to make enough space, the new flagellin will fail to enter the channel. In summary, the pushing force from the apparatus can only push a distance of 56 nm, but it can push multiple flagellin monomers, up to as many as all the flagellin monomers within *LS*×56 nm, if their presence in the vestibule of the channel restricts the insertion of the new flagellin.

C) About the estimate of flagellin length of 56 nm, we assume that the subunits are partially unfolded mainly as α-helices (Stern and Berg, 2013) but not completely unfolded and this is further clarified in our revised manuscript (subsection “Estimation of *V. alginolyticus* flagellin length” in Material and methods).

*Subsection “Simulating flagellar growth by a one-dimensional single-file diffusion model”, point 1: LS as described is really equivalent to a loading length in units of 56 nm – better to quote this length directly so that it can easily be compared to the filament length in Figure 4. Naively I'd expect that the occupancy saturates and the rate slows when the filament is longer than the loading length. But it seems that this happens at filament lengths longer than the loading length. Discuss/explain.*

A) We thank the reviewers for the suggestion. We modified Figure 4 and added the real length of LS in the figure legend.

B) It is indeed true that the occupancy saturates and rate decays at the flagellar length longer than the loading strength. This is because diffusion of flagellin must be taken into account. When the flagellum is short, the flagellins can easily reach the growing tip by diffusion even beyond the loading strength, which effectively extends the loading strength. That is why the decay of growth rate occurs at longer flagellar length.

*Subsection “Simulating flagellar growth by a one-dimensional single-file diffusion model”, point 3: Is it realistic that (both?) touching units stop? Rather than diffusing more slowly as a larger object?*

In our simulation we assume that the units stop because the following reasons. First, this is the low Reynolds number environment, the inertia force is neglectable and this is not elastic collision condition. Secondly, the simulation runs with a very small time step to ensure the stability of simulation under the equation of motion. Thirdly, there is no knowledge of the binding/unbinding rate of flagellin units. To simplify this, we assume the units are not binding to each other. Even though the touching units stop at this time step, in the next time step, they can move independently if there are enough space.

*Subsection “Model parameter space search and the best fit”, last sentence: Figures do not show that there is a dependence of growth rate in the limit of long filaments upon "secretion force" (LS?). If this has been demonstrated, it should be shown. It's not clear that the model is expected to predict this, especially in the light of the previous comment which seems to preclude monomers pushing each other. But perhaps in fact stopping upon contact DOES provide a mechanism for pushing, as hitting injected monomers will ratchet the Brownian Motion of those further in.*

Our previous 3-part piecewise linear fitting was not accurate and as suggested we have decided to remove it from Figure 2. The growth rate decays when flagellar length is longer and does not converge to a plateau. We have changed our text accordingly.

*3) Subsection “Fluorescent labeling of flagellar sheath for real time monitoring of flagellar growth”, third paragraph: discuss/explain/fix the conflict between statements that flagella are anchored to the surface and the assumption that they are 3D helices. Are they helices anchored at lowest points? Is there any evidence for this, e.g. 3D imaging? The 2D-3D correction is minor to the point of being unnecessary, but also it is wrong if filaments are fully anchored to the surface and thus in fact 2D.*

We thank the reviewer for raising this question and also for suggestions on how to improve our presentation. In our experiments, flagella are not fully anchored to the surface. A 3D image of *Vibrio alginolyticus* polar flagellum taken by the SIM was added to Figure 2—figure supplement 1, showing that flagella are 3D helices, with the lowest points attaching to the surface.